Quantification of water requirement of some major crops under semi-arid climate in Turkey

Aydın Yusuf yaydin@gantep.edu.tr
Nizip Vocational Training School Department of Plant and Animal Production Organic Farming Programme Samandöken Village-Nizip/GAZİANTEP, Gaziantep University , Nizip , Gaziantep , Turkey
Hossain Mohammad Anwar
Electronic publication date: 2022 Jul 7
Publication date: 2022
Volume: 10
Electronic Location ID: e13696
Received 2022 Feb 25; Accepted 2022 Jun 16
Copyright: ©2022 Aydın
Copyright year: 2022
Copyright holder: Aydın
License: This is an open access article distributed under the terms of the Creative Commons Attribution License, which permits unrestricted use, distribution, reproduction and adaptation in any medium and for any purpose provided that it is properly attributed. For attribution, the original author(s), title, publication source (PeerJ) and either DOI or URL of the article must be cited.
License URL: https://creativecommons.org/licenses/by/4.0/

Keywords: Crop water requirement, Cropwat 8.0, Reference evapotranspiration, CLIMWAT 2.0, Irrigation scheduling, Pistachio, Olive, Almond, Grape

Funding: The author received no funding for this work.

==============================
The Southeastern Anatolian Region of Turkey is located in semi arid climate zone and therefore requires an efficient water use. Well-planned irrigation with optimum water required by the crops is essential for the limited water resources of the region. The numerical tool CROPWAT of the Food and Agriculture Organization (FAO) was used for modelling efficient irrigation of local crops pistachio, olive, almond and grape without reducing the yield. Local climatic, soil, plant and rainfall information were used as inputs to CROPWAT model to predict the reference evapotranspiration (ETo) values. The crop water requirement (CWR) for pistachio, olive, almond, and grape was calculated as 1,294.0 mm, 659.4 mm, 790.2 mm, and 752.0 mm, respectively, The number of irrigation needed during growth stages was determined as eight for pistachio, three for olive, six for almond and five for grape.

Introduction

The accurate estimation of the crop evapotranspiration (ETc) is vital in the management and development of water resources. The scarcity of fresh water resources or the misuse of water in agriculture has made the efficient and profitable use of water even more important. The Gaziantep province has a transition climate characteristic between a Mediterranean and continental climate. Therefore, the rainfall regime of the Mediterranean climate zone differs in the region, and most of the rainfall occurs in the winter months. In addition, the evaporation, which increases with the high temperature in the summer months, increases the ETc and the need for irrigation water (Aydın, 2021). Effective and efficient use of water resources requires a good planning, development and operating system and cannot be considered independent of the vegetation pattern of the region. Accurate estimation or determining the ETc and amount of irrigation water for the plants in the crop pattern enable to develop appropriate irrigation programs and can also prevent irrigation-induced yield reductions. The main purpose of irrigation is to ensure continuity in ETc and to provide sufficient moisture in the root zone of the crops under less rainfed conditions. Since water is an important input for crop production, it should be well-planned and distributed effectively. The expected benefit from irrigation can be increased and irrigation efficiencies can be improved with the appropriate modifications of conventional irrigation systems (Balasaheb & Sudarsan Biswal, 2020). Water resources should be operated at an optimum level in terms of irrigation parameters such as ETc, crop water requirement (CWR) and net irrigation water (NIR). The calculation of CWR is very important in regional balancing of water resource, correct planning and operation of water resources, determination of possible water demands that may occur in the future, increasing irrigation efficiency, operating irrigation systems and maintaining water supply–demand balance. ETc and CWR values vary depending on climate, planting area, planting time, soil type (Sharma & Tare, 2021) and can be estimated by using direct or indirect measurement methods and meteorological data. The direct measuring method of ETc is often difficult and costly. Therefore the ETo can be calculated using the Penman-Monteith equation in the windows-based CROPWAT 8.0 software developed by Food and Agriculture Organization (FAO) of the World. Factor Kc is used to calculate ETc from ETo. In addition, estimations of irrigation water requirements and yield decreases in response to critical reductions in irrigation water allow improvement of irrigation programs (Bhat et al., 2017). The irrigation water requirement for different crop patterns can be calculated with this model (using soil, plant and climate data), and reliable estimates can be carried out.

The FAO-CROPWAT model has been successfully used in many parts of the world for different crop patterns under irrigated as well as rainfed conditions (Alhassan et al., 2015). Many studies have been conducted to determine the crop evapotranspiration (ETc) and crop water requirement (CWR) for different plants in different climate and soil conditions. The CROPWAT 8.0 software has been widely used in different geographical regions and climatic conditions to calculate ETc and irrigation water requirements (IWR) of field crops (Mehta & Pandey, 2016; Veeranna & Mishra, 2017; Bhat et al., 2017), vegetables (Memon & Jamsa, 2018) and rice (Nithya, Shreedhara & Shivapur, 2015; Mehta & Pandey, 2016; Verma et al., 2019) in Egyptian territories, (Gabr, 2021a; Gabr, 2021b; Sharma & Tare, 2021), and also of Jatropha curcas L. which is a tree or shrub used for biofuel production (Moseki, Murray-Hudson & Kashe, 2019).

The model was preferred due to reliable results reported in studies conducted in semi-arid climatic conditions such as Gaziantep region, without causing a decrease in yield or under controlled reduction conditions. The aim of this study was to determine the ETc and crop water requirements (CWR) for pistachio, olive, grape and almond (which are the main crops of Gaziantep province), to determine appropriate irrigation programs and the priority order in irrigation planning.

Materials & Methods

Study area and data handling

Study area

The study was carried out in the Gaziantep provinces of the Southeastern Anatolia Region. The Gaziantep province is located between 36°28′ and 38°0′ east longitudes and 36°38′ and 37°32′ north latitudes (Fig. 1).

Figure 1 Gaziantep province and its locational view in Turkey.

Gaziantep province has the characteristics of transition climate between Mediterranean and the continental climate. The winter season is cold and rainy, while the summer months are quite hot and dry. The highest and lowest temperatures of the year vary between 44 °C and −17.5 °C, and the annual average temperature is 15.1 °C. Long-term annual average rainfall is 463 mm of which most rainfall occurs in winter. The rainfall is quite low in the summer season when evapotranspiration is high and the plant water requirement is at the most (Aydın, 2004).

Total land area of Gaziantep province is 682,280 ha, and 50.6% (345 415 ha) of the lands consists of agricultural lands. Orchards have the largest share (62%) in agricultural lands, followed by field crops (35%) and vegetables fields (3%). Pistachio has the largest share (65%) in orchards, followed by olive (21%), vineyard (8%) and other fruit (6%) growing lands (Gaziantep Directorate of Provincial Agriculture and Forestry, 2021). Therefore, pistachio, olive, grape and almond cultivation has an important place in the regional economy (Table 1). The province ranks the first in the coverage area of pistachio orchards and the amount of pistachio production in Turkey (Aslan, 2017).

Table 1 Data for four crops cultivated in Gaziantep Province.

Crops	Acreages, (he)	Production (ton)	Average yield (kg tree−1)	Number of trees	Source	
Pistachio	133,539a	75,298	4.2b	21,982,745	Aslan (2017) and bAydın (2004)	
Olive	44,646a	54,632	6	10,273,943	National Olive and Olive Oil Council of Turkey (2021)	
Grape	16,239a	125,928a	–	–	a Gaziantep Directorate of Provincial Agriculture and Forestry (2021)	
Almond	116.6	3,629	19	230,140	Yılmaz & Okay (2021)	
Notes.

a Values have been taken from Gaziantep Directorate of Provincial Agriculture and Forestry (2021).

b Dry red shell fruit yield value.

Description of CROPWAT 8.0 model

The CROPWAT 8.0 model is a windows-based software used to determine ETc, developed by the contributions of the Food and Agricultural Organization (FAO), Land and Water Development Division of Italy, Irrigation and Development Studies of Southampton and Egypt National Water Research Centre (Balasaheb & Sudarsan Biswal, 2020). The model allows to carry out simulations under water stress conditions and to estimate possible yield decrease in response to a decrease in irrigation water (Verma et al., 2019). The model is effectively used as a decision support system in irrigation project design studies and in the development of irrigation programs due to ease of use. Likewise, this model is used in different ecological conditions to estimate the evapotranspiration rate, to determine irrigation requirements and deficit irrigation for different crops (Moseki, Murray-Hudson & Kashe, 2019). In addition to the calculation of crop water requirement, the model can be used to prepare irrigation programs under different irrigation conditions and different plant growth conditions, to develop more efficient irrigation methods, and to evaluate crop yield under rainfed conditions or adequate irrigation conditions (Kartal et al., 2019).

The agronomists, irrigation and water resources engineers engaged in agricultural production can also use the same model to calculate ETo (Mehta & Pandey, 2015). The aforementioned model considers the grass as the reference plant, which does not have water restrictions, fully covers the ground surface and has good growth. ETc values of the crop are calculated by using Eq. (2) where the ETo values are multiplied by the factor Kc for the local crops under study. ETo values are derived by the CROPWAT 8.0 model based on the Penman-Monteith method given in Eq. (1).

Data requirements

The CROPWAT 8.0 model requires four basic data groups including climate parameters (maximum, minimum and average temperatures, relative humidity, sunshine, wind speed), rainfall, soil (heavy, moderate, light), and the characteristics of the crop studied. The climate data required for the model can be obtained using the CLIMWAT 2.0 software, which includes the climate data measured from different parts of the world, as well as the data of local meteorology stations (Ewaid, Abed & Al-Ansari, 2019).

Other parameters, required for the CROPWAT 8.0 model, like crop coefficient (Kc), yield response factor (Ky), crop growth period values provided in Tables 2–3 for pistachio were compiled from local studies (Aydın, 2004). In addition to aforementioned properties, others such as rooting depth and critical depletion fraction (P) for olive, almond and grape plants were manually compiled from FAO56 and other related literature (Food and Agriculture Organization, 2002 and Ministry of Food, Agriculture and Livestock of Turkey, 2017). The climate and soil parameters of Gaziantep province including rainfall and soil properties and the data on the plant provided in Tables 2 and 3, were used in the CROPWAT and CLIMWAT softwares to calculate the ETo and CWR values,

Table 2 Data for the four crops in the study.

					Crop growth periods (days)a		
Crops	Scientific name	Planting and harvesting date	Critical depletion fraction (p)	Rooting Dept.b (cm)	Initial	Crop development	Mid -season	Late season	Total growing period, (days)	
Pistachio	Pistacia vera L.	15 Mar–25 Oct	0.40	120–150	20	69	80	60	229	
Olive (oil)	Olea europaea	15 Mar–25 Oct	0.65	90–150	30	80	55	80	245	
Grape (Table)	Vitis vinifera	15 Mar–25 Oct	0.35	50–120	22	54	83	65	224	
Almond	Prunus Amygdalus	15 Mar–25 Oct	0.40	60–120	28	50	95	50	223	
Notes.

a Food and Agriculture Organization (2002), p: 165.

b Ministry of Food, Agriculture and Livestock of Turkey (2017).

Table 3 Crop coefficients for four crops in the study.

Crop type	Kc coefficients	
	Kc-ini	Kc-mid	Kc-end	
Pistachioa	0.42	1.51	0.39	
Olive (oiler)	0.66	0.72	0.68	
Grape (Table)	0.63	0.86	0.43	
Almond	0.40	0.90	0.61	
Notes.

a Aydın (2004).

The agricultural soils of Gaziantep Province have a medium-heavy texture (Kalkancı& Şimşek, 2020); thus, soil texture for the agricultural soils in this study were considered as medium textured in the solution of the CROPWAT software.

Crop evapotranspiration (ETc)

The actual ETc values of the crops studied are calculated by the use of two parameters namely the ETo and the factor crop coefficient (Kc) as given in Eq. (2). ETo values were calculated bythe CROPWAT 8.0 model using the Penman-Monteith equation (Allen et al., 1998). Most parameters in the FAO56-Penman-Monteith model can be measured directly, while some parameters can be calculated using related equations of the subjects. (1) ETo−PM=0.408Δ∗Rn−G+γ900Tmean+273u2es−eaΔ+γ1+0.34u2

where; ETo is the reference crop evapotranspiration (mm d−1); Rn is the net radiation (MJ m−2), G is the soil heat flux (MJm−2), Tmean is the average air temperature (°C), U2 is the wind speed at 2 m height (m s−1), es is the saturation vapor pressure (kPa), Δ is the slope of vapor pressure curve (kPa °C−1), γ is the psychrometric constant (kPa °C−1) (2) ETc=Kc×ETo.

The coefficient Kc, used to determine ETc, is the result of combined effects of transpiration from the plant surface and evaporation from the soil surface and reflects the common characteristics of both surfaces (Shah, Suryanarayana & Parekh, 2017).

Crop coefficient (Kc)

The crop coefficient (Kc) is defined as the ratio of the ETc to the ETo (Food and Agriculture Organization, 2002). The FAO-56 model is based on the concepts of ETo and crop coefficient (Poblete-Echeverría & Ortega-Farıas, 2013). Four important and different periods of crops are taken into account in the calculations of Kc based on ETc. (i) The initial stage is the period from seed sowing to germination and 10% ground coverage; (ii) the growth period is between the end of the initial stage and the period when the crop completes the growth and the percentage of ground cover reaches 70–80%; (iii) mid-term is the stage when crop growth is completed or the crop fully grows and matures with the onset of flowering; (iv) the period between the maturation stage and the harvest is considered the end period (Çetin, 2013).

The Kc coefficients were used to calculate ETc values via ETo values based on the FAO56-PM equation (Eq. 1). The Kc values of local crops pistachio (Aydın, 2004), olive, grape and almond (Ministry of Food, Agriculture and Livestock of Turkey, 2017), were compiled from the literature considering the plant growth stages (initial, mid, end stages).

Crop water requirement (CWR)

Semi-arid climate zone characteristics of the region in mind, pistachio, olive, grape and almond are the main agricultural and economical crops of the Southeastern Anatolia Region of Turkey and require an efficient irrigation program. The CWR, including the losses and the ETc, the amount of net irrigation water requirement (NIRs) and irrigation schedules were determined for the same crops by using CROPWAT 8.0 software. The inputs of the model are long-term annual average rainfall and effective rainfall values both obtained by the USSC method, and other soil and plant parameters like Kc, Ky etc. The monthly ETo values calculated with the CROPWAT model using Gaziantep climate data are given in Table 4.

Table 4 Climate characteristics, rainfall and ETo of Gaziantep Province obtained using the CLIMWAT tool attached to the CROPWAT software.

Month	Min Temp	Max Temp	Humidity	Wind	Sun	Rad	ETo	Rain	Eff rain	
	°C	°C	%	km/day	hours	MJ/m2/day	mm/day	mm	mm	
January	−0.8	7.3	79	121	2.5	6.5	0.82	99	83.3	
February	0.4	9.2	73	130	2.5	8	1.14	77	67.5	
March	3.1	14	64	138	5.4	13.8	2.08	79	69.0	
April	7.2	19.7	59	138	7.2	18.7	3.25	52	47.7	
May	11.7	25.7	51	130	9.8	23.9	4.66	34	32.2	
June	16.7	31.2	43	181	12.1	27.7	6.5	6	5.9	
July	20.8	35.3	37	199	12.5	28	7.52	2	2.0	
August	20.6	35.1	37	164	11.5	25.1	6.67	1	1.0	
September	15.9	31.2	41	121	10	20.3	4.78	2	2.0	
October	9.8	24.2	50	78	7.5	14.1	2.67	37	34.8	
November	4.4	16.4	68	78	4.9	8.9	1.42	63	56.6	
December	1.3	9.9	76	104	2.3	5.7	0.91	98	82.6	
Average	9.3	21.6	56	132	7.3	16.7	3.54	550	484.7	

Irrigation scheduling and net irrigation requirement

The expected benefit from irrigation can be obtained only if the amount of water needed by the crops and proper irrigation program are determined in advance. The irrigation schedules of pistachio, olive, grape and almond are shown in Figs. 2–5. The net irrigation water requirement (NIR) of crops is the amount of water required for crops to fully develop or to bring the decreased available soil water content in the plant root zone back to the field capacity. The NIR varies with the crop pattern and climate characteristics of the region.

Figure 2 Irrigation schedules for pistachio.

Figure 3 Irrigation schedules for olive.

Figure 4 Irrigation schedules for grape.

Figure 5 Irrigation schedules for almond.

Net irrigation water requirements are calculated using the following equation; (3) NIR=ETc−Peff

where; ETc is the crop evapotranspiration, and Peff is the effective rainfall.

The CROPWAT model uses the water budget technique detailed by Allen et al. (1998) to determine irrigation water demand (IR). For this purpose, the following equation (Eq. 4) is used in IR calculation. (4) Dc=Dp+ETc−P−I+SRO+DP.

where: Dc: current day’s soil water deficit (net irrigation requirement) in the root zone; Dp: previous day soil water deficit; ETc: current day crop evatranspiration rate; P: actual gross precipitation; I: amount of final irrigation injected into the field today; SRO: water loss by runoff from the soil surface, DP: deep percolation (Sharma & Tare, 2021).

The amount of water to be diverted from the source to the irrigation area is calculated by the ratio of NIR to the irrigation efficiency. Total irrigation efficiency is one of the most essential factors in determining irrigation modules (flow, l/sec/ha) in irrigation projects.

During the transmission of water from the source to the plant root zone in the irrigation area, some part of water is lost through mechanisms of transmission losses, deep percolation, infiltration and surface runoff. In addition to the aforementioned losses, some of irrigation water is needed for soil leaching, sowing and planting. Therefore, water losses must be taken into account and the total irrigation efficiency must be calculated to apply the amount of net irrigation water to the root zone and to bring the water content to the field capacity.

In the irrigation process, there are some definitions like RAM&TAM readily availably moisture (RAM) and total available moisture (TAM), etc. RAM is the amount of water that plants can easily take in the root zone without encountering any water stress. TAM expresses the total amount of moisture contained in the soil around plant root zone. The RAM value is also defined as a certain percentage of the TAM.

Therefore, in irrigation applications, water reductions in the root zone are allowed until the RAM level, and after this point, irrigation is started and recycled until the root zone moisture level reaches to the field capacity. TAM is also defined as the difference between soil moisture content (SMC) at field capacity and permanent wilting point (PWP) by Kanber (2015).

Effective rainfall (Peff)

In arid or semi-arid climatic conditions, agricultural production is highly dependent on the rainfall. However, the full amount of rainfall is not always usefull for crops use. Some of rainfall is lost through deep infiltration and surface flow. Moseki, Murray-Hudson & Kashe (2019) explains that as the amount of water available to plants after all the losses have taken place is referred to as the effective rainfall. The estimation of effective rainfall values depend on the location of the study area and the amount of rainfall occurs in the region. The Eqs. (5) and (6) below, recommended for CROPWAT 8.0 by the US Soil Conservation Service, were used to determine the monthly effective rainfall values (Balasaheb & Sudarsan Biswal, 2020).

(5) Peff=P∗125−0.2∗P125forP≤250mm

(6) Peff=125+0.1∗PforP>250mm.

where, Peff is the effective rainfall in mm, and P is the total rainfall in mm.

Effective rainfall values are calculated by using Eqs. (5) and (6) based on the total rainfall values. In cases or periods where effective rainfall values are higher than the ETc values, there is no need for any irrigation. Therefore, CWR values are calculated based on the cases where ETc values are higher than effective rainfall values.

Results

The parameters such as ETo, CWR, effective rainfall and total irrigation water requirement for the crops studied were calculated using the CROPWAT model. The climate characteristics of Gaziantep Province calculated with CROPWAT are given in Table 4, and the other outputs of the model are given in Tables 5–9 and Figs. 2–5.

Table 5 Crop water requirement for pistachio.

Month	Decade	Stagea	Kc	ETc	ETc	Eff rain	Irr. Req.	
			coeff	mm/day	mm/dec	mm/dec	mm/dec	
March	2	Init	0.42	0.87	5.2	14.4	0.0	
March	3	Init	0.42	1.04	11.4	21.3	0.0	
April	1	Deve	0.46	1.33	13.3	18.1	0.0	
April	2	Deve	0.62	2.01	20.1	15.6	4.5	
April	3	Deve	0.78	2.9	29.0	14.0	15.0	
May	1	Deve	0.94	3.94	39.4	12.7	26.6	
May	2	Deve	1.1	5.12	51.2	11.2	40.1	
May	3	Deve	1.27	6.69	73.6	8.1	65.5	
June	1	Deve	1.44	8.46	84.6	4.2	80.3	
June	2	Mid	1.52	9.91	99.1	1.0	98.2	
June	3	Mid	1.52	10.43	104.3	0.9	103.4	
July	1	Mid	1.52	11.15	111.5	1.0	110.5	
July	2	Mid	1.52	11.77	117.7	0.6	117.2	
July	3	Mid	1.52	11.23	123.6	0.5	123.1	
August	1	Mid	1.52	10.7	107.0	0.4	106.5	
August	2	Mid	1.52	10.31	103.1	0.3	102.8	
August	3	Late	1.52	9.29	102.2	0.4	101.8	
September	1	Late	1.4	7.58	75.8	0.1	75.7	
September	2	Late	1.21	5.8	58.0	0.0	58.0	
September	3	Late	1.02	4.17	41.7	2.0	39.7	
October	1	Late	0.83	2.82	28.2	8.3	19.9	
October	2	Late	0.65	1.72	17.2	12.1	5.1	
October	3	Late	0.47	1.05	9.4	11.8	0.0	
					1,426.7	159	1,293.9	
Notes.

a Init, initinal; Deve, development; Eff. Rain, effective rain; Irr. Req, Irrigation requirements.

Table 6 Crop water requirement for olive.

Month	Decade	Stage	Kc	ETc	ETc	Eff rain	Irr. Req.	
			coeff	mm/day	mm/dec	mm/dec	mm/dec	
March	2	Init	0.66	1.37	8.2	14.4	0.0	
March	3	Init	0.66	1.63	17.9	21.3	0.0	
April	1	Init	0.66	1.89	18.9	18.1	0.8	
April	2	Deve	0.66	2.15	21.5	15.6	5.9	
April	3	Deve	0.67	2.5	25	14.0	11.0	
May	1	Deve	0.68	2.85	28.5	12.7	15.8	
May	2	Deve	0.69	3.21	32.1	11.2	21.0	
May	3	Deve	0.70	3.69	40.6	8.1	32.4	
June	1	Deve	0.71	4.17	41.7	4.2	37.5	
June	2	Deve	0.72	4.67	46.7	1.0	45.7	
June	3	Deve	0.73	4.97	49.7	0.9	48.9	
July	1	Mid	0.73	5.36	53.6	1.0	52.6	
July	2	Mid	0.73	5.66	56.6	0.6	56.0	
July	3	Mid	0.73	5.4	59.4	0.5	58.9	
August	1	Mid	0.73	5.14	51.4	0.4	51.0	
August	2	Mid	0.73	4.96	49.6	0.3	49.3	
August	3	Late	0.73	4.47	49.1	0.4	48.7	
September	1	Late	0.72	3.91	39.1	0.1	39.1	
September	2	Late	0.71	3.42	34.2	0.0	34.2	
September	3	Late	0.71	2.88	28.8	2.0	26.8	
October	1	Late	0.7	2.35	23.5	8.3	15.2	
October	2	Late	0.69	1.84	18.4	12.1	6.3	
October	3	Late	0.68	1.53	16.8	14.4	2.5	
November	1	Late	0.67	1.23	12.3	16.4	0.0	
November	2	Late	0.66	0.94	3.8	7.5	0.0	
					827.4	185.5	659.4	

Table 7 Crop water requirement for grape.

Month	Decade	Stage	Kc	ETc	ETc	Eff rain	Irr. Req.	
			coeff	mm/day	mm/dec	mm/dec	mm/dec	
March	2	Init	0.63	1.3	7.9	14.4	0.0	
March	3	Init	0.63	1.6	17.1	21.3	0.0	
April	1	Deve	0.64	1.8	18.2	18.1	0.1	
April	2	Deve	0.68	2.2	22.0	15.6	6.4	
April	3	Deve	0.72	2.7	26.8	14.0	12.8	
May	1	Deve	0.77	3.2	32.1	12.7	19.4	
May	2	Deve	0.81	3.8	37.8	11.2	26.6	
May	3	Mid	0.86	4.5	49.7	8.1	41.6	
June	1	Mid	0.87	5.1	51.3	4.2	47.1	
June	2	Mid	0.87	5.7	56.7	1.0	55.7	
June	3	Mid	0.87	6.0	59.7	0.9	58.8	
July	1	Mid	0.87	6.4	63.8	1.0	62.8	
July	2	Mid	0.87	6.7	67.3	0.6	66.8	
July	3	Mid	0.87	6.4	70.7	0.5	70.2	
August	1	Mid	0.87	6.1	61.2	0.4	60.8	
August	2	Mid	0.87	5.9	59.0	0.3	58.7	
August	3	Late	0.83	5.1	55.8	0.4	55.4	
September	1	Late	0.76	4.1	41.1	0.1	41.0	
September	2	Late	0.69	3.3	33.1	0.0	33.1	
September	3	Late	0.62	2.5	25.4	2.0	23.4	
October	1	Late	0.56	1.9	18.8	8.3	10.5	
October	2	Late	0.49	1.3	13.0	12.1	0.9	
October	3	Late	0.44	1.0	4.0	5.2	0.0	
					892.4	152.5	752	

Table 8 Crop water requirement for almond.

Month	Decade	Stage	Kc	ETc	ETc	Eff rain	Irr. Req.	
			coeff	mm/day	mm/dec	mm/dec	mm/dec	
March	2	Init	0.40	0.83	5.0	14.4	0.0	
March	3	Init	0.40	0.99	10.9	21.3	0.0	
April	1	Init	0.40	1.14	11.4	18.1	0.0	
April	2	Deve	0.45	1.45	14.5	15.6	0.0	
April	3	Deve	0.55	2.04	20.4	14.0	6.4	
May	1	Deve	0.65	2.73	27.3	12.7	14.5	
May	2	Deve	0.75	3.51	35.1	11.2	24.0	
May	3	Deve	0.86	4.54	50.0	8.1	41.9	
June	1	Mid	0.91	5.38	53.8	4.2	49.5	
June	2	Mid	0.91	5.94	59.4	1.0	58.4	
June	3	Mid	0.91	6.25	62.5	0.9	61.6	
July	1	Mid	0.91	6.68	66.8	1.0	65.8	
July	2	Mid	0.91	7.05	70.5	0.6	70.0	
July	3	Mid	0.91	6.73	74.0	0.5	73.6	
August	1	Mid	0.91	6.41	64.1	0.4	63.7	
August	2	Mid	0.91	6.18	61.8	0.3	61.5	
August	3	Mid	0.91	5.57	61.3	0.4	60.9	
September	1	Late	0.9	4.84	48.4	0.1	48.4	
September	2	Late	0.83	3.99	39.9	0.0	39.9	
September	3	Late	0.77	3.15	31.5	2.0	29.4	
October	1	Late	0.71	2.39	23.9	8.3	15.6	
October	2	Late	0.64	1.72	17.2	12.1	5.1	
October	3	Late	0.60	1.36	4.1	3.9	0.0	
					913.8	151.2	790.2	

Table 9 Irrigation schedules for pistachio, olive, grape and almond.

Crops	Date	Day	Stage	Rain	Ks	Eta	Depl	Net Irr	Deficit	Loss	Gr. Irr	Flow	
				mm	fract.	%	%	mm	mm	mm	mm	l/s/ha	
Pistachio	3 Jun	81	Deve	2.2	1	100	41	175.9	0	0	251.2	0.36	
22 Jun	100	Mid	0	1	100	40	176.1	0	0	251.6	1.53	
9 Jul	117	Mid	0	1	100	42	182.4	0	0	260.5	1.77	
25 Jul	133	Mid	0	1	100	42	184.2	0	0	263.2	1.90	
11 Aug	150	Mid	0	1	100	42	184.0	0	0	262.9	1.79	
29 Aug	168	Mid	0	1	100	40	175.7	0	0	251.0	1.61	
26 Sep	196	End	0	1	100	41	176.3	0	0	251.9	1.04	
29 Oct	End	End	0	1	0	8	39.3					
Olive	12 Jul	120	Mid	0	1	100	66	285.2	0	0	407.4	0.39	
8 Sep	178	End	0	1	100	65	284.1	0	0	405.9	0.81	
14 Nov	End	End	0	1	0	18	90.0					
Grape	10 Jun	88	Mid	0	1	100	36	156.7	0	0	223.9	0.29	
6 Jul	114	Mid	0	1	100	35	152.4	0	0	217.7	0.97	
30 Jul	138	Mid	0	1	100	36	156.0	0	0	222.9	1.08	
26 Aug	165	End	0	1	100	36	156.1	0	0	223.0	0.96	
24 Oct	End	End	0	1	100	30	130.8					
Almond	11 Jun	89	Mid	0	1	100	42	144.9	0	0	207.1	0.27	
4 Jul	112	Mid	0	1	100	40	140.4	0	0	200.6	1.01	
25 Jul	133	Mid	0	1	100	41	142.9	0	0	204.2	1.13	
16 Aug	155	Mid	0	1	100	40	140.7	0	0	201.1	1.06	
12 Sep	182	End	0	1	100	41	142.0	0	0	202.8	0.87	
23 Oct	End	End	7.7	1	100	22	84.0					

Estimates of reference evapotranspiration and effective rainfall

The monthly average ETo values were quite low (0.82 mm d−1) in winter months, and reached the highest value (7.52 mm d−1) especially in July with the increase in maximum temperatures and accordingly evaporation. Low rainfall and high evaporation in summer months causes a decrease in the soil moisture content, and the need for irrigation becomes more evident in this period. The differences in ETo values can be associated with the transitional climate of Gaziantep Province between the temperate Mediterranean and the colder continental climate. ETc also increases in dry seasons under high air temperature, high wind speed and low relative humidity conditions (Ewaid, Abed & Al-Ansari, 2019).

Crop water requirements (CWR) of pistachio, olive, grape and almond

The CWR values for pistachio, olive, grape and almond plants are given in Tables 5–8. The ETc values were calculated considering the growth stages of plants. The ETc values increased from the initial stage to the growth stage, but the ETc values decreased towards the last stage. The differences in ETc values are due to the difference in the Kc coefficients that reflect the characteristics of the crops (Table 3). The value of coefficient Kc is not constant throughout the crop growth stages, therefore cause differences in the seasonal irrigation water requirement of the crops. Similar differences are observed in the effective rainfall values due to the irregular rainfall regime. All rainfall values (throughout the year) cannot be used to determine the CWR. The amounts of effective rainfalls are very low during the maximum growth periods of the crops (Tables 5–8). Therefore, the amount of effective rainfalls, in arid and semi-arid climate regions, can be neglected in CWR calculations to prevent possible risks in crop growth (Kanber, 2015). In this study, only the total amount of the annual effective rain (provided in Table 4 calculated using Eqs. (4) and (5) as 484.7 mm), was used in the calculation of CWR during the vegetation period of the studied plants.

In addition to climate parameters, many factors such as soil structure, genetic characteristics of crops, geographic location, and conventional or modern agricultural technologies used have a direct influence on ETc. Therefore, the CWR values of the crops investigated in this study were significantly different between the growth stages of the plants. The CWR values based on 10-day interval are listed as:

pistachio (1293.9 mm) > almond (790.2 mm) > grape (752 mm) > olive (659.4 mm) and provided also in Tables 5–8.

The irrigation schedules of pistachio, olive, grape and almond are shown in Table 9 and Figs. 2–5.

The flow values in the last column of Table 9 are the irrigation module values calculated for the crops studied. The irrigation module is calculated by considering the highest ETc value. The calculated peak module is used to determine the channel capacities in irrigation systems (Kanber, 2015). Peak modulus values for pistachio, grape and almond were calculated as 1.9, 1.08 and 1.13 l/sec/ha, respectively, in the last decade of July, and 0.81 l/sec/ha for olive in the first decade of September. The amount of water to be diverted from the source and the diversion times in irrigation programs can be easily determined by using the findings of this study for Gaziantep Province where water resources are limited and insufficient.

Net irrigation water requirements (NIR) and gross irrigation water requirements for full irrigations calculated by Eq. (3) changed between 179.2 and 256.0 mm for pistachio, 284.7 and 406.7 mm for olive, 155.3 and 221.9 mm for grape, and 142.2 and 203.2 mm for almond. The number of irrigation application determined in irrigation program was 8 times for pistachio, 3 times for olive, 5 times for grape and 6 times for almond. Irrigation starting times were determined as 3rd of June for pistachio, and 12th, 10th and 11th of June for olive, grape and almond, respectively.

Discussion

The literature review showed that the irrigation program based on ETc studies have not previously been carried out for olive, grape and almond in province of Gaziantep. Whereas, the number of irrigation and frequency of irrigation for Gemlik olive variety, were reported as 10 times and 21-day intervals in neighbouring province Şanlıurfa (Anlağan Taş et al., 2016). The total irrigation water requirement of Gemlik variety was determined as 749 mm and irrigation starting and ending dates were determined as the first half of April and the second half of October (Anlağan Taş et al., 2016). The number of irrigation, frequency of irrigation and the amount of total irrigation stated for Şanlıurfa Province are quite different from Gaziantep Province. The differences can be attributed to the fact that the Gemlik olive variety is more sensitive to water deficiency, as well as the different climatic characteristics of Gaziantep & Şanlıurfa provinces. A similar condition has been observed for grape. In Tarsus-Adana conditions, the number of irrigation for grape was determined as 10–11, the total irrigation water was between 599 and 671 mm, and the irrigation was recommended between second half of May and second half of July (Kanber et al., 2017). In the present study, the results revealed that irrigation should be applied five times between June and October and total amount of irrigation water was determined as 752 mm,. The differences in the amount of irrigation, the number of irrigation and the irrigation period can be attributed to the differences in climate, altitude and geographic locations of Gaziantep & Adana provinces. A limited number of studies have been found on pistachio irrigation using different irrigation methods. While Kanber, Önder & Köksal (1992) determined the ETc value of pistachio as 803 mm in the 20-days irrigation interval, Bilgel, Dağdeviren & Nacar (1999) calculated ETc value as 841 mm in the 15-days irrigation interval. Surface irrigation methods were used in both studies. In another study, conducted by Aydın (2004) on pistachios irrigated with drip irrigation, the ETc value was determined as 813 mm for 20 days irrigation interval and it is reported that irrigation of pistachios in Gaziantep province should start at the beginning of June. The findings of Aydın (2004) are in accordance with the starting time of irrigation determined in the present study.

Conclusions

In the present study, the main objectives were to determine both the amounts of ETc values and the crop water requirement (CWR) values for four different crops (pistachio, almond, grape and olive) of the Southeastern Anatolian Region of Turkey, in paricular Gaziantep Province.

The CROPWAT 8.0 model of FAO was used to determine ETc and CWR values. The input parameters to the model such as climate and rainfall parameters are provided by the CLIMWAT 2.0 program.

According to the results obtained from the model, in a fixed period of duration (from March to November), CWR values were determined as 1,294 mm for pistachios, 790.2 mm for almonds, 752.0 mm for grapes and 659.4 mm for olives. Therefore, the order of crop irrigation priority follows as: pistachio>almond>grape>olive.

The higher the CWR values, the more frequent irrigation is required: pistachio, eight times; almonds, six times; grapes, five times; and olive, three times from March to November.

Based on the data provided and the fact that harvesting time of the three crops (pistachio, grape and almond) is the summer time, it is decided that the most critical crop in Gaziantep Province is the pistachio regarding irrigation requirements.

Supplemental Information

Supplemental Information 1 Cropwat solution file for Pistachion

Click here for additional data file.

Supplemental Information 2 Cropwat solution file for Olive

Click here for additional data file.

Supplemental Information 3 Cropwat solution file for Grape

Click here for additional data file.

Supplemental Information 4 Cropwat solution file for Almond

Click here for additional data file.

Supplemental Information 5 Pistachio parameters compiled from this thesis

The crop coefficients and yield response factor and some other parameters related with pistachio were taken local studies and also from this PhD thesis.

Click here for additional data file.

Supplemental Information 6 Climatic data for Gaziantep Province used for Cropwat taken from Climwat

Click here for additional data file.

Supplemental Information 7 Climate data of Gaziantep Provinces for Cropwat taken from Climwat

Click here for additional data file.

Supplemental Information 8 Climate data of Gaziantep Provinces for Cropwat taken from Climwat

Click here for additional data file.

Supplemental Information 9 Climate data of Gaziantep Provinces for Cropwat taken from Climwat

Click here for additional data file.

The author thanks Melih Akay, Teknotar Irrigation Company Manager in Gaziantep, for his technical support and Prof. Nihat Yildirim from Gaziantep University for editing the text.

Additional Information and Declarations

Competing Interests

Author Contributions

Data Availability

The author declares there are no competing interests.

Yusuf Aydın conceived and designed the experiments, performed the experiments, analyzed the data, prepared figures and/or tables, authored or reviewed drafts of the article, and approved the final draft.

The following information was supplied regarding data availability:

The raw data from local research and literature used to calculate crop water requirements and irrigation scheduling in the study were available in the Supplemental Files.

The software used in this study is available at FAO: https://www.fao.org/land-water/databases-and-software/cropwat/en/.

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
