# Peer review of "Quantification of water requirement of some major crops under semi-arid climate in Turkey"

_PeerJ, doi:10.7717/peerj.13696_

## Round 0.1 · original submission · Major Revisions

Dear Authors, You need substantial improvement in your manuscript. Please revise according to the reviewers' comments.

Reviewer 1 ·

Basic reporting

There is still huge scope to improve the article structure, figures, tables, conclusions and discussion. Please see "4. Additional comments".

Experimental design

All the materials and methods used in the study have not been described sufficiently in the M&M section. The uncertainties in the data and procedure should be assessed and discussed. Please see "4. Additional comments".

Validity of the findings

The final output of the study "number of irrigation and total depth of irrigation" was qualitatively compared with some reported values. More quantitative comparison is needed to validate the findings. Please see "4. Additional comments".

Additional comments

General comments
The paper entitled ‘‘Comparison of reference evapotranspiration and crop water requirement of some major crops under semi-arid conditions in Turkey’’ describes the quantification and assessment of water requirement and irrigation scheduling for some major crops grown in Turkey. The topic is of interest to the readers of the journal. However, a thorough revision is needed from the Title to Conclusions section. The authors have used many Tables and Figures, but I think they can present their results more precisely. What I am missing is the quantitative corroboration of their findings against some observations. The authors can consider the following specific comments for revising the manuscript.

Specific comments
Title
“Comparison of Reference Evapotranspiration and Crop Water Requirement of Some Major Crops under Semi-Arid Conditions in Türkiye” can be shortened like: “Quantification of Water Requirement of Some Major Crops under Semi-Arid Climate in Turkey”.

Abstract
Line16-17: “--excessive use of water—“ is not the only problem. Please rewrite the sentence.
Line-18: Write “can be” instead of “will be”.
Line 20-22: Rewrite the sentence. It does not make sense.
Line 26-33: The results of rainfall, crop water requirement and irrigation are not meaningful. Where does the CWR of pistachio (1294 mm) come from? You have only 484.7 mm rainfall and 176.3 mm irrigation, so these are not enough to meet the CWR. Check your values.

Keyword
Do not repeat words in Keyword and Title.

Introduction
The arguments in the Introduction has not been well developed. The importance of the work should be elaborated based on recent literature so that readers can understand the novelty of this research. The limitations and scope of this methods should be discussed.

Materials and Methods
Line 83: Use a small but clear map. The text on the map is not legible.
Line 100: What do you mean by “crops breading” in Table 1. Please use only SI unit.
Line 103-115: Please use the appropriate terminologies throughout the manuscript.
Line 118-131: Very clearly mention the data type and source of each data. Also describe how you processed all the data before the model-incorporation.
Line 124: All the values mentioned here are not given in Table 2.
Line 133: Define a term at it first appearance, such as Critical Depletion Fraction. Is it soil moisture depletion factor?
Line 161: “plant water consumption (ETc)” ETc is actual evapotranspiration, which is equal to the net crop water requirement. Plant water consumption can be less than the “theoretical ETc (I mean, if you calculate ETc = ET0 *Kc)” in any stress condition. Please use the terminologies recommended in agricultural water management discipline.
Line 161-174: Justify that you have used the right Kc values. It can vary geographically. You have discussed 4 growth stages, but you have three values in Table 3. You have not referred Table 3 in the text.
Line 179-186: Rainfall during the non-crop period may not be considered effective (You did it correctly). Please mention the procedure in M&M section. Describe all your methods in M&M section but NOT in Results section.

Results
Line 190-192: These are not results. These are Materials and Methods, so place them accordingly.
Line 199: “---model are given in the following tables and figures” Please write specifically with the Table numbers.
Line 201: Write “Estimated” instead of “Estimation”.
Line 202-206: These are not results. These are Materials and Methods, so place them accordingly.
Line 255-256: The extra effective rainfall in one decade will be stored in the root zone soil and will be available for the plants in the subsequent decades. Why did not you consider this in calculating irrigation requirement (Table 5-8)?
Line 291-296: These are not results. These are Materials and Methods, so place them accordingly.
You have used many Tables and Figures, and I think some of the information have been repeated. Try to reduce the number of Figure or Table.

Discussion
Line 300-320: Try to improve the discussion part with the underlying science of your results and recent literature. The final output of the study "number of irrigation and total depth of irrigation" was qualitatively compared with some reported values, but more quantitative comparison is needed to validate the findings.

Conclusions
Line 323-337: Rewrite your conclusions including only your main findings, final conclusions and/or recommendations
Line 323-329: These are Materials and Methods, so remove them.

Figures and Tables
Figure 2-5: Use smaller Figures but large and clear legends. Please add the irrigation dates on the Figures. Write in full form-----, such as “Readily available moisture”----------------etc.
Table 2: Write the date appropriately.

·

Basic reporting

The essence needs improvement. I suggest that you remove the repeating sentences between lines 20 - 26. The authors should refer to the following research paper
1)Managing irrigation needs using the FAO-CROPWAT 8.0 model: a case study from Egypt
2) Modeling of pure irrigation water requirements using FAO-CROPWAT 8.0 and CLIMEWAT 2.0: the Tina Plain and East South Alcantara regions, North Sinai , A Case Study of Egypt

The literature review and introduction need more detail with the latest research paper. I suggest author to improve the description from lines 64 - 68. For the introduction, you should refer to and quote "Assessment of Irrigation Requirement and Scheduling under Canal Command Area of ​​Upper Ganga Canal using Cropwater Model".

The author needs to define the research gap.

Experimental design

The author needs to include CWR and irrigation scheduling in the methodology section instead of the results section.

Validity of the findings

(Lines 71-74): The research objective is to estimate the CWR but in the results section, only the NIR calculation is shown. The author needs to include field application losses to calculate CWR. It would be more informative if the author provided the results of monthly water requirements for all crops. It would also be helpful to estimate the canal capacity (as shown in the conclusion)

---

## Round 0.2 · Minor Revisions

Dear authors, thank you for the revised manuscript. Though one reviewer endorsed the review, another reviewer raise a question about English quality of the manuscript. It would nice of you polish the English of the manuscript and submit the final version of the manuscript.

Reviewer 1 ·

Basic reporting

Please follow the academic writing convention/style of the journal. You can abbreviate a term (such as crop evapotranspiration) in the first appearance and then use only the abbreviated form (such as ETc) throughout.

Line 280: “cropss”??? “Kc coefficients” Is there more than one Kc coefficient? Maybe--- many values of this coefficient. Please edit very carefully the whole manuscript to remove all such typos and language errors.

Experimental design

The author has tried to improve the manuscript, but another round of thorough revision is needed.

Validity of the findings

The validation of your modeling results is still poor. Since you do not have any measured values, please try to compare all your main results with the findings of previous studies.

Additional comments

Abstract
Rewrite the abstract. Write about the importance and objectives first and then the materials and methods and finally your main findings. Your study is not about water quality, right? Do not stress on irrelevant issues.
Introduction
Please follow the academic writing convention/style of the journal. You can abbreviate a term (such as crop evapotranspiration) in the first appearance and then use only the abbreviated form (such as ETc) throughout.
Results and discussion
Line262-263: This is not a nice way of referring all the tables/Figures in one place. Refer them in specific places when needed.
Line 280: “cropss”??? “Kc coefficients” Is there more than one Kc coefficient? Maybe--- many values of this coefficient. Please edit very carefully the whole manuscript to remove all such typos and language errors.
Line 283-284: “Effective rainfall is the amount of rainfall used to determine the CWR and is the remain of water from deep percolation and surface runoff.” Such very basic definition-type statements should be in the Materials and Methods section.

Validation of your modeling results is still poor. Since you do not have any measured values, please try to compare all your main results with the findings of previous studies.

Conclusions
The aim of this study was to determine the crop evapotranspiration (ETc) and crop water requirements (CWR) for pistachio, olive, grape, and almond to determine the appropriate irrigation programs and to determine the priority order in irrigation planning. Therefore, in your conclusions section, try to answer all these issues very clearly with numerical values. Do not refer to any Table or Figure in this section.

·

Basic reporting

no comment

Experimental design

no comment

Validity of the findings

no comment

Additional comments

All issues are addressed. Manuscript is quite good now and can be accepted for publication after the small modification
(1) Remove “Crop evapotranspiration (ETc) in estimating”(Line 163)
(2) Line (167-69) remove. These are repeating sentences.
(3) Equation of Irrigation scheduling is missing (methodology section)

---

## Round 0.3 · accepted · Accept

The revised manuscript can be accepted for publication

The Section Editor provided an annotated PDF (attached) and said:
> Suggested edits occurred on lines: 15, 18, 20, 22, 42, 74, 76, 89, 109, 148, 227, 229, 239, 243, 329, 330, 331, 358, 361, 373, 374.